# For Learning Analytics to Be Sustainable under GDPR—Consequences and Way Forward

Thashmee Karunaratne 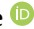



Department of Computer and Systems Sciences, Stockholm University, SE-146 07 Kista, Sweden; thasmee@dsv.su.se

**Abstract:** Personalized learning is one of the main focuses in 21st-century education, and Learning Analytics (LA) has been recognized as a supportive tool for enhancing personalization. Meanwhile, the General Data Protection Regulations (GDPR), which concern the protection of personal data, came into effect in 2018. However, contemporary research lacks the essential knowledge of how and in which ways the presence of GDPR influence LA research and practices. Hence, this study intends to examine the requirements for sustaining LA under the light of GDPR. According to the study outcomes, the legal obligations for LA could be simplified to data anonymization with consequences of limitations to personalized interventions, one of the powers of LA. Explicit consent from the data subjects (students) prior to any data processing is mandatory under GDPR. The consent agreements must include the purpose, types of data, and how, when and where the data is processed. Moreover, transparency of the complete process of storing, retrieving, and analysing data as well as how the results are used should be explicitly documented in LA applications. The need for academic institutions to have specific regulations for supporting LA is emphasized. Regulations for sharing data with third parties is left as a further extension of this study.

**Keywords:** learning analytics; general data protection regulations; GDPR; policy; data processing; student consent

## 1. Introduction

Massive volumes of data about students and their learning environments are being accumulated in various education databases due to the use of digital tools. Learning analytics (LA) emerged as a promising approach that uses these data to support students to achieve a better learning experience [1,2]. The primary goal of LA is to enhance learning based on the data about learners and the environments learning occurs and thereby maximise the academic performance of the learners [1]. Assessing students' academic progress [3], predicting student performance [4], and identifying any potential issues affecting student progress in education [1] are to name a few of the common LA practices. Based on the insights and the predictions LA delivers, teachers and educational institutions can make decisions to tailor the education services for a maximised benefit to individual learners from an early stage and at a near-real time [5].

The quality of LA depends heavily on the data available for processing. In other words, the quality of data is an essential factor for the efficiency and effectiveness of LA [2]. Some of the LA applications use data collected directly from students through surveys and other data collection methods. These data are typically collected after the consent of the data subjects (individuals the data belongs to—students). However, there exist other types of data that are scraped, retrieved, or harvested from educational technology (edTach) systems, which are used to support the delivery of education. In using these data in LA applications, it needs to be collected, pre-processed, refined, curated and restructured according to the need of the LA applications. This activity is typically carried out by data controllers (individuals or entities who process data) [6]. However, these data, which might also contain sensitive or personal data, may not necessarily be accumulated and processed

after the explicit consent of the data subjects [7]. Furthermore, many of these data are in the form of digital traces or digital footprints the learners leave behind when they use edTach systems and are automatically archived in data repositories. Therefore, the users may not necessarily be aware of where, how, and in what forms the data are stored and used [8].

The General Data Protection Regulation (GDPR) (EU 2016/679, 2016), regulates the collection, storing and use of personal data. The provisions and the limitations of LA are heavily dependent on the GDPR legislation of management and use of data for the benefit of the data subjects (students) [9]. According to the literature, LA may use students' personal and sensitive data [9], mainly when the LA outcomes are used for providing personalized support to the learners [6]. Hence, naturally, the point of departure for effective LA is a careful investigation of how the GDPR guidelines on fair and lawful processing of data prohibit or encourage LA approaches. In principle, the GDPR clauses related to (1) explicit consent from users on processing their personal data, (2) data minimalization, (3) processing data leading to profiling user groups, (4) aggregating data from different data sources, (5) curate data for obtaining new insights, (6) anonymization of personal data, (7) data ownership, (8) transparency of decision making, (9) right to opt-in and out are of concern when performing LA practices. Therefore, in the light of the legal grounds sets by GDPR, it is essential to examine the boundaries of LA, which may have not necessarily been addressed explicitly in the current knowledge scope of the subject [6–8,10]. In recent studies such as [11,12], the importance of the recognition of the privacy of data subjects in LA applications is highlighted. Followed by the mentions of such needs in the LA literature, the study presented in this article aims at exploring the possibilities for GDPR friendly Learning Analytics by closely analysing the contemporary LA research practices under the GDPR concerns. Therefore, it tries to answer the questions of: (1) What current practices of LA are reported in the literature? (2) What are the GDPR provisions and limitations on data needed by LA tasks? (3) What implications arise when LA meets the GDPR conditions? (4) What is the way forward for lawful LA under the effect of GDPR? This study excludes, however, the ethical aspects and consequences of LA, although it is a critical and essential focus not necessarily falling into the frame of legal concerns. Therefore, this study merely emphasises the legal argumentation of the use of data for LA activities.

The rest of the article is organised as follows. The next section briefs the methodology base for the study. Section 3 captures an overall review of LA practices, where a pre-hypothesis is derived such that personal data might be an integral part of "effective" LA, where learners can individually and collectively benefit from the decisions made for an enhanced learning experience. Section 4 briefs the GDPR framework and scans through the GDPR clauses relevant for LA, emphasising the essential obligations by LA to be GDPR friendly. Section 5 summarise the process towards LA that complies with the GDPR. Finally, Section 6 bundles the outcome of the study with the main findings and potential further extensions.

## 2. Methodological Approach

As stated in the preceding section, the aim of this research is, first, to systematically investigate the current landscape of LA research. A partial systematic study was performed in this regard, with two stages (1) automated text analysis on LA article corpus to check if the articles include terms related to the legal (and ethical) aspects of data, and (2) summarise the focusses of contemporary LA. Second, content analysis is conducted for exploring the GDPR concerns limiting lawful LA. Based on the outcome of the analysis, a way forward for GDPR friendly LA is proposed.

### 2.1. Systematic Literature Study (SLR)
2.1.1. Automated Text Analysis

A systematic review of the current literature (SLR) is typically anticipated to place the exact cornerstones of the current LA landscape, since it may help to determine how far the topics concerning the relevance of sensitive data and/or methods are evolved in the

literature. However, in the recent past (here a time window of 2017–2020 is considered), around 150 SLR studies have been published regarding different specific focuses of LA, according to the databases ACM, Google Scholar, EBSCO, IEEE, Scopus, and Web of Science. Therefore, instead of merely adding one more to the collection of SLR studies, this study proposes an automated SLR that is a partial systematic review. On the other hand, due to the sheer number of relevant articles, a typical SLR, with a strong protocol of inclusions and exclusion to reduce the size of the corpus may have the chance of excluding impotent articles that do not penetrate through the inclusion and exclusion criteria. Therefore, this study followed the approach of automating the analysis of the selected article corpus and presenting the outcomes by means of text visualization. This approach makes sure that the important articles significant to obtain a snapshot of the current status of LA are not excluded. Hence, the automated review performs data mining on peer-reviewed LA articles from the above-mentioned databases and automated processing of them using text mining methods, as described in the subsequent sections.

The following research protocol presented in Figure 1 was applied to finding relevant articles.

```
In Title{{learning}AND {analytics}}
        AND
        {{student}AND {data}} OR {{learner}AND {data}}
        Limiters:
                Peer reviewed
                Date Published: 20170101–20201231
                Language: English
```

**Figure 1.** The SLR article search protocol.

The search query is simple and only includes the term Learning analytics in the title and student data in the body text as the contextual boundaries. A time window of three years (2017–2020) is the temporal boundary. Using text mining methods, (1) the titles, (2) abstracts and (3) the related keywords are retrieved. These three datasets are processed and presented as word clouds. The idea of creating three separate word clouds is to make sure the LA terms that are used in each of the articles are not missed. The word clouds for titles abstracts and keywords are created using the algorithm presented in Figure 2.

```
BEGIN
{ READ fileset
CORPUS {
        REMOVE PUNCTUATION;
        REMOVENUMBERS;
        REMOVEWORDS,  stopwords("english")),  ("learning",  "analytics",
        "students", "student", "data","education", "using","higher","study"))
        }
CREATE Termdocument matrix
CREATE wordcloud
PLOT wordcloud
}
END
```

**Figure 2.** Text mining algorithm for creating the document cloud.

The texts are processed using software R [13], with standard text mining techniques including data cleaning, stop words removal, stemming and ranking the words. For the abstracts, the top 200 words above frequency 10 are plotted in the word cloud. For titles and keywords, it was the word set more frequent than 5.

2.1.2. A comparison Document Corpus

In the above-mentioned protocol for creating the LA corpus, articles that do not mention "Learning Analytics" in the title are excluded irrespective of how relevant the article is, and hence the argument of the automated approach is diluted. However, this limitation was unavoidable, even in an automated SLR, given the size of the corpus resulted from including Learning Analytics in "anywhere" in the articles. However, arguably, such a strong exclusion might result in a drift in the LA landscape since the LA focuses of the articles that are not using "Learning Analytics" in the title will naturally be opted out. To examine if such drift exists, this study used a controlled set of articles from an LA publication archive; that is the proceedings of the Learning analytics conferences (LAK). In this set, articles that have "Learning analytics" anywhere in the article is included. In developing the LAK corpus same algorithm in Figure 2 is used with the same parameters.

*2.2. Content Analysis Methods for Exploring GDPR*

Content analysis is in demand when a qualitative study requires systematic exploring of contents (or documents). This study follows a methodology of content analysis to closely analyse the clauses of the GDPR framework that applies to LA concerns. All the references for the GDPR and its Recitals are taken from EU 2016/679, 2016 [14]. The GDPR clauses considered in this paper are not exhaustive in any means, but inclusive, pointing out the essential relevance for lawful LA under GDPR.

## 3. The Current Landscape of Learning Analytics
*3.1. A Systematic Survey on the Current Status of the Filed*

LA research has been in its hype during the past few years. Over one million hits in Google Scholar for the term proves the popularity of the field. Leading research databases; ACM, Google Scholar, EBSCO, IEEE, ERC, Scopus, and Web of Science lists over 6500 research articles with Learning Analytics in the title. The Learning Analytics conference (LAK) has produced 267 research publications since 2011. Hence, as described in Section 2, two search protocols we applied; (1) global search for "Learning Analytics" in the title, and (2) "Learning Analytics" anywhere in the articles from LAK.

The initial search retrieved 638 articles. After the screening for duplicates, language, and non-peer-reviewed, 402 relevant hits remained in the corpus. This corpus is referred to as LA corpus hereafter. The LAK article set, after screening for inclusion-exclusion, resulted in 151 articles. This article set could create meaningful visualizations using all the terms present in each article. Word clouds of titles of LAK articles are created for side-by-side comparison with the word cloud of the titles of the LA corpus. Figure 3 shows these two-word clouds.

As shown in Figure 3, similar words stand out in both clouds. It is not surprising to have terms such as *online* and *performance* as common words since most LA methods are based on data in the edTech tools [5], and possibly focusing on student performance [15]. Similarly, *predictions*, *model building*, *assessment data*, *student engagement*, *achievement*, etc. have also been among frequent topics. The term *multimodal* also has been used by many publications. Therefore, it could also be understood that data from multiple sources were of interest.

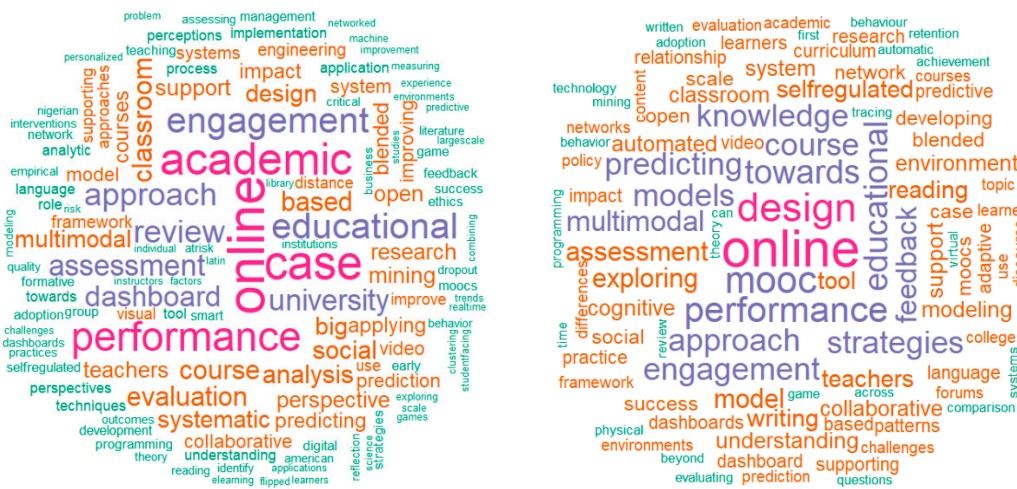

**Figure 3.** Titles of LA publications from 2017–2020: article corpus with LA in title (**left**) and LAK articles (**right**).

Abstracts and keywords of the LA corpus: To obtain a deeper understanding of the contents concerning the research questions and significant outcomes, a word cloud of the abstracts of the 402 articles is created. Furthermore, the keywords of the same 402 articles are compiled to a third word cloud. Figure 4 presents these clouds.

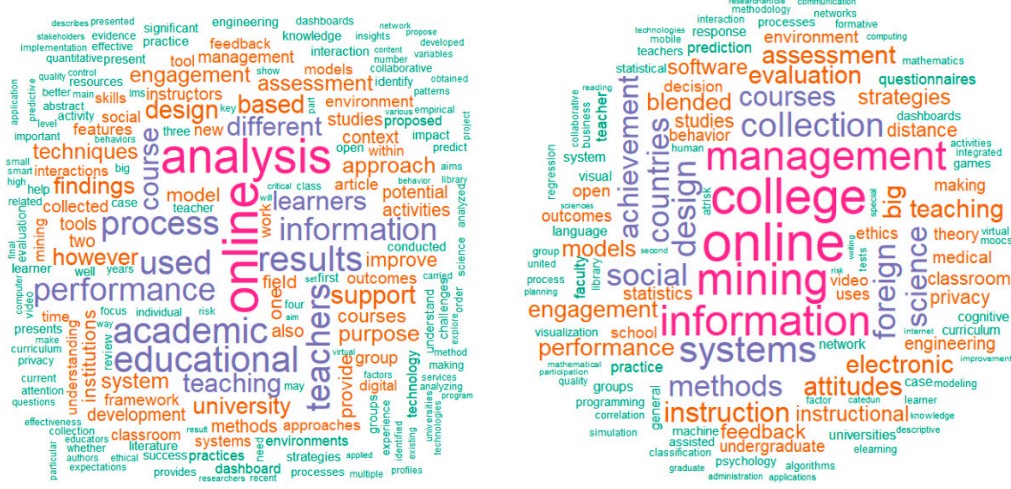

**Figure 4.** LA publications from 2017–2020: abstracts (**left**) and keywords (**right**).

As the clouds of titles, these two clouds in Figure 4 include the term "online" as the most frequent term. For the keywords, *online learning* could be the main area, as well as "data mining". Therefore, these results indicate that the LA field prioritizes "online" settings and data. "Student engagement", "learning process", "academic performance" are the main discussion points in the abstracts. "Data mining", "information management", "systems design" etc. stand out as keywords. However, there are no terms related to the lawful and ethical use of data that appeared in any of the clouds, although plenty of articles in the corpus considers predictive modelling and student behaviour data.

Hence, the outcome of this analysis shows no evidence concerning the legal (and ethical) aspects of processing personal data, although personal data is used in many of the previous LA studies. As a consequence, there is a need for asking the question of whether the current LA landscape is explicit and transparent on lawful management of data that is required by GDPR.

*3.2. Data Sources for LA*

LA approaches are grounded on data. "LA initiatives are data-hungry pushing institutions toward new and expanding data collection activities" as reported by [16]. According to [17], "the most common types of data collected for the practices were students' academic behaviours and their background information". A study in [18] lists different stakeholders of LA and the sources of data those stakeholders are interested in. Accordingly, learning and course management systems (LMS, CMS), student information systems (SIS), virtual learning environments (VLE), personalized plug-ins including third-party digital tools, other digital platforms for online/offline exams and assessment, enterprise reporting platforms (ERP), business intelligence platforms, and third-party administrative systems and other historical data repositories including surveys are prospective data sources for LA [18]. With a focus on the student's perceptions of the privacy of their data, [19] lists many primary sources of data for LA including, individual characteristics of students such as student's prior knowledge and academic performances, learning activities, interactions with the learning environment (learning pathways, downloads), the curricular benchmarks such as the learning outcomes and the historical information of courses, as well as peer-to-peer and student-teacher interactions including social network activities. In a review of LA to support teacher inquiry, [20] introduced a set of categories of data such as engagement in learning activities, assessment score, engagement in discussion activities, engagement with educational resources/tools, demographics, and student behaviour.

Almost all the LA research utilizes data from one or more sources mentioned above. Among the sources, data from intelligent tutoring or learning management systems are the most popular [21]. The prevalence of automatically accumulated information-rich data in these systems could be the reason for the popularity. Interpreting learning deficits as a health issue, [22] presents a data model that requires micro-, intermediate and macro-level interventions in curing the learning flaws of students. A quite broader and logically distinct grouping of data has been presented in [23], categorizing data into three groups, namely directed, automated and volunteered. Although the reference to data in [23] is more general, the same concept can be applied in the education setting as well. The "directed data" is collected by tracking tools, such as surveillance cameras, GPS instruments, and other behaviour detection tools. Tracking student behaviour in classrooms by recording teaching-learning sessions, or group activities, to understand the student dynamics are among the standard practices [24]. "Automated data" includes data residues at digital platforms due to the use of that platform. Log data LMS data, clickstreams, as well as data from the use of social media, are treated as automated data. The majority of LA approaches use this type of data [25].

In contrast, volunteered data are generated purposely by the users. Information provided about the learning such as feedback questionnaires, contents developed such as the assignments, group works, contents in communication threads and discussion forums, other knowledge contributions, and information generated by participating in surveys can be stated as some examples of such data [23]. Works of [26] justify that "LA systems allow educational institutions to track individual student engagement attainment and progression and even wellbeing in near real-time".

Multimodal LA is capturing the attention of the LA field due to the benefits of data aggregation. Multiple stakeholders, sophisticated data architectures, different data owners' sharing common LA goals and sharing data to build LA models play key roles in multimodal LA [27]. Current LA architectures use multimodal tools such as xAPI (learning experience trackers), and aggregate data from sources such as physiological (wellbeing) data and digital traces (log data of behaviour of individuals in learning platforms), student records and multimedia data [27]. However, this article also excludes the review of ethical and legal grounds of multimodal LA.

In summary, the following conclusions can be drawn based on the current status of LA.

- Impactful LA requires sophisticated data models, which may not necessarily be filtering from a single data source.
- LA methods demand personal and sensitive data. The data sources such as the LMS, click logs, personal information about students (demographics), and health and wellbeing data of students contain sensitive information.
- Transparency of data management: explicit mention of how sensitive data is managed, including the clearance needed by ethical and lawful use prior to processing is not a mandatory point in the current LA.

*3.3. Key Constructs of LA*

LA literature provides approaches ranging from different complexities with respect to data and methods. Data summaries, visualization, perspective analytics, predictive modelling and forecasting are the key methodologies that could be applied on data from a single source to an aggregation of data from many sources. However, the current landscape of LA inclines towards more automatic and real-time methods comprising data from a variety of sources ranging from simple static data tables to real-time behaviour capturing sensor data accumulators [28]. Accordingly, monitoring individual students, predicting the behaviour of individuals in the learning environment, etc. have been recognized as important for impactful LA. The study [29] reviews the current scope of practices in visual LA and elaborates on the multifaceted use of data for providing evidence in decision making. Monitoring student behaviour in real-time in learning environments, providing insights into students learning paths, identifying interaction patterns of individuals with teachers, peers and learning contents, dropout prediction, early intervention, has thereby been identified as key strengths of LA. It also further claims that the majority of the tools are "more often designed for instructors than for students" [29].

Another recent systematic study [30] argues the gaps in prioritizing students in LA solutions "failures to ground analytics matrices to learner goals and expectations" which need to be bridged in by through sophisticated dynamic and data-driven methods extending from digital footprints to "more aspects that characterize learners, their needs, and expectations". Refs [29,30] mention the importance of preserving the trust among students when implementing LA for increased effectiveness and impact. Building LA models with standard (simple) analytics methods and sophisticated data architectures with aggregated data are also shown as trendy and impactful [27]. Standard approaches, thereby, are statistical analysis methods, such as descriptive, inferential, and multivariate analyses, and simple machine learning algorithms.

When referring to the effectiveness of LA, a frequent claim is the power and ability of LA to provide personalized support for individuals based on the evidence aggregated from multiple data sources [26]. As it further exemplifies, the models for interventions should also include external parameters such as student wellbeing as predictor variables for increased accuracy in clustering students to different groups. Recommendations from this study to the LA community are mainly concentrated around predictive modelling and the need for evidence-based predictive elements when forecasting if students are potential dropouts, at-risk or successful. Although the study never recommends the legal and ethical grounds to count on in designing and evaluating LA, it ultimately concludes by pointing out the essential need of evaluating the performance of the LA models, based on the interventions in practice in academic environments.

The main characteristics of the current LA practices concentrate around educational institutions and are based on individual courses or program segments covering a broad range of disciplines as [17] concludes. Accordingly, the LA main purposes must be for facilitating learning and academic progress, enhancing the effectiveness of learning support, improving administrative efficiency and, supporting to attain institutional goals [17]. Accumulating evidence from the field, this study further proposes the need of examining "potential relationships between contextual factors such as size and location of institutions,

purposes and outcomes of learning practices as well as the factors for data-informed culture among institutions" [17].

In summary, the following conclusions can be drawn from the above discussion.

- Autonomous and near-real-time analytics methods are in demand for impactful LA.
- Methods on characterizing individual students and grouping as successful or at risk (clustering) for personalized support are promoted.
- Based on data architectures aggregating from multiple sources, with multi-stakeholder involvement, LA approaches potentially provide more meaningful analytics results.

### 3.4. Data Subjects vs. Data Controllers Dilemma on Privacy and LA

Although the ethical debate for the data subjects vs. data controllers dilemma is of high and equal relevance, this study focuses only on the legal aspect of privacy concerns of LA. Preserving privacy of the data subjects while performing LA has been recognized in principle embraced by previous research [9,16,18,19,31–33]. However, as [9] points out, the complexity of the field of education due to different legal obligations and rights of stakeholders could hinder the effective application of legal measures in real-life situations. The speed, scale, and power of the tools, and the increased use of different sensors capturing the student dynamics in learning add fuel to this complexity as mentioned by [34]. Furthermore, the students might not necessarily be aware of this prevalence in their educational environments, compared to their environments of social media. Focussing on the same line [33] argue on a "privacy paradox", where students unwillingly uncover sensitive data under the factors such as fear of inferiority, lack of awareness and the sense of consequences, which may be practically advantageous for intuitions when implementing LA [33]. According to [18] the governance body of the student data typically are the educational institutions. The centralized decision-making power of the governance body usually allow to authorizes how the data is being used for the benefit of the effective functioning of the institution, which may not necessarily cohere with the benefits of the students (data subjects) [18]. Furthermore, the encouragement of data sharing among stakeholders for increased impact is constantly pointed out in the LA field [27] but a similar level of concern is yet to develop on the lawful and ethical ways of sharing data.

On another side of the debate lies the arguments of human centeredness in succeeding LA approaches, such as participatory design, co-creating solutions together with all stakeholders and users (learners), and user-centred design [35]. Thereby, users gain more control over managing their data, shifting the power of governance of data towards serving the user. The study [35] however acknowledge that "in terms of the ethics of LA, and growing concerns about the misuse of data, human-centred design has the potential to shift LA from something done to learners toward something done with learners" [35].

In summary, the following observations on data ownership and use could be outlined from this discussion:

- The complexity of LA settings hinders the effective application of privacy measures to data subjects.
- Data sharing among stakeholders improves the impact and usability of LA solutions.
- User-centric approaches of LA allowing the users to control over their data are still in their infancy.

### 3.5. Recap of the Current LA Scope

The aim of this section was to capture the overall landscape of LA. Arguably, abundant studies related to building predictive models for grouping and supporting individuals exist among LA literature. Furthermore, evidence of how student privacy is maintained during data processing—or explicit disclaimers on if personal data is used and how—are missing from the methodology descriptions of these articles. LA applications, in general, should aim at benefiting the learners, but observations are that the current impacts incline more towards benefiting the academic institutions to showcase the success stories of the academic bodies. Thus, the current LA landscape brings in evidence to ring an alarm on

how important it is to regulate the need for transparency and explicit on the legal and ethical aspects for lawful LA practices under GDPR, since in the current shape, this fact is given the least priority.

If these conclusions to be mapped to the concerns of GDPR clauses presented in the previous section, i.e., (1) explicit consent from users on processing their personal data, (2) data minimalisation, (3) processing data leading to profiling user groups, (4) aggregating data from different data sources, (5) curation of data for obtaining new insights, (6) anonymization of personal data, (7) data ownership, (8) transparency of decision making, (9) right to opt-in and out, are of concern when performing LA practices, first, the exact meaning and exceptions of the GDPR clauses should be understood.

## 4. The GDPR Framework for Lawful Management of Personal and Sensitive Data

The EU regulation 2016/679 [14] concerns the protection of natural persons regarding the processing of personal data. The free movement of such data is commonly referred to as the General Data Protection Regulation (GDPR). The regulation was in force since May 2018. The direct relevance of GDPR for LA is Article 6 [14], which focuses on the lawfulness of the processing of personal data. As summarised in Section 2 above, LA may use students' personal and sensitive data. Therefore, students, as natural persons, may require protecting themselves with regards to processing their personal data. Based on the abovementioned literature review findings of the LA methods and practices, there is a need for carefully examining the lawful ways of processing data for LA purposes.

### 4.1. The Starting Point: Interest of the Data Subject

According to Recital 46 of GDPR, the genuine interest of the data subject (students), i.e., "*it is necessary to protect an interest which is essential for the life of the data subject or that of another natural person*" should be the primary concern for processing student personal data. The freedom of the data subject to decide to be included/excluded in the event of data processing is ensured in Recital 2. The legal obligations should closely be followed when obtaining consent from data subjects. This applies to events of, anonymizing of sensitive and personal data, the transparency of the LA process, aggregating data from multiple sources, grouping (automatic)—profiling—students under different categories, control over data including opt-in and out and empowering the data subject for managing their personal data.

### 4.2. Informed Consent

Article 6:1(a) refers to obtaining informed consent from individual data subjects prior to processing their data. The consent should be explicit, specifically stating the exact data collected and purpose as well as how the data are processed and how the outcome is used. Furthermore, Recital 40 further elaborates on the authorizations for the consent of personal processing data. Thereby, "*consent of the data subject concerned or some other legitimate basis*" should be secured by "*including the necessity for compliance with the legal obligation to which the controller is subject or the necessity for the performance of a contract to which the data subject is party or in order to take steps at the request of the data subject prior to entering into a contract*" according to Recital 40.

LA concerns: Obtaining informed consent from data subjects when data is collected directly from them, such as in survey questionnaires and other methods collecting volunteer data is a well-established and mandatory practice [36]. However, when the data processing follows data-driven methods, and specifically if the data are automatic, this step might not be explicitly followed [37]. It may also be challenging to meet the required level of being explicit in such processing since data-driven methods allow the algorithms to find what data is useful to be processed. Thus, the fact that students must consent to a *particular purpose of data processing* could possibly become complicated under these methods. In other words, it may not simply be possible to be specific in identifying each individual purpose of a given LA study prior to the collection of data. If an institution does not know what

exact personal data are needed, it becomes naturally challenging to inform the data subject and ask for consent before the collection of data [9]. In such a situation, general consent to data analysis/analytics would be possible [38]. However, this could be a limitation from the abovementioned legal perspective. In other words, defining the purpose of data processing as "to perform Learning Analytics" is insufficient from a legal perspective. The details of the purpose, data and processing must be provided to the data subject in the consent form.

### 4.3. Right to Be Forgotten

Article 7:3 concerns the possibility to withdraw a previously given consent by a data subject: "*The data subject shall have the right to withdraw his or her consent at any time. The withdrawal of consent shall not affect the lawfulness of processing based on consent before its withdrawal. Prior to giving consent, the data subject shall be informed thereof. It shall be as easy to withdraw as to give consent*". Recital 33 mentions "*It is often not possible to fully identify the purpose of personal data processing for scientific research purposes at the time of data collection*" which is often the case with LA, and therefore, "*data subjects should be allowed to give their consent to certain areas of scientific research when in keeping with recognized ethical standards for scientific research. Data subjects should have the opportunity to give their consent only to certain areas of research or parts of research projects to the extent allowed by the intended purpose.*" (Recital 33).

The right to be forgotten is further strengthened by Article 17:1(b), such that erasing personal data should be conducted without further delay if the data is no longer of interest of the academic institution for serving the purpose defined, or, the data subject request for withdrawal of the consent. Under 17:1(e) "*Personal data also have to be erased for compliance with a legal obligation in Union or Member State law to which the controller is subject*". Recital 66 further specifies the right to erasure in the online environment; as "*a controller who has made the personal data public should be obliged to inform the controllers which are processing such personal data to erase any links to or copies or replications of those personal data*". However, as Recital 65 states, "*further retention of the personal data should be lawful where it is necessary, for exercising the right of freedom of expression and information, for compliance with a legal obligation, for the performance of a task carried out in the public interest or in the exercise of official authority*".

LA concerns: According to Recital 33, processing data should not be performed if the exact data requirement and the outcomes of the processing cannot be estimated prior to data processing. As summarised in the preceding section, many LA approaches use predictive modelling where the data and outcome are not predictable before executing the algorithms—so as the reason to refer to these methods as data-driven. Furthermore, when a dataset is pre-processed (collected and curated to be used by algorithms) it may not leave track of the original data element, due to many reasons including the computational and storage costs. It might be extremely difficult in general to remove any data point in a later stage, unless the whole dataset is dropped from the analysis/results.

Moreover, with Article 65, GDPR is restricting the actions of preserving data for the purpose of future LA, without explicitly and adequately mentioning the exact purpose when the consent is obtained from students. Thus, ultimately, data processing without keeping the record of original data as well as preserving (saving) or continuous collection of data for future purposes is limited by these two GDPR clauses.

### 4.4. Access Rights

Pursuant to Article 15, a data subject may request access to any or all the personal data an institution has processed that relates to themselves. The institution is bound to offer "*confirmation as to whether or not personal data concerning him or her are being processed, and, where that is the case, access to the personal data*" (Article 15(1)). Furthermore, the data subject has the right to know, and when required, to request information regarding the purposes of the processing and the categories of personal data concerned, "*the recipients or categories of recipient to whom the personal data have been or will be disclosed, in particular recipients in third*

*countries or international organizations*; and, "*where possible, the envisaged period for which the personal data will be stored, or, if not possible, the criteria used to determine that period*"; as well as " *the existence of the right to request for rectification or erasure of personal data or restriction of processing of personal data concerning the data subject or to object to such processing; and "the right to lodge a complaint with a supervisory authority*". Moreover, in Article 15, "*if the personal data are not collected from the data subject,* he or she may have the right to request "*any available information as to their source*". In the event of the automated processing of data, the subject has the right to know "*the logic involved in any automatic personal data processing*" under Recital 63. A copy of all the personal data must be provided within a month from the requested date (Recital 61).

### 4.5. The Role of Anonymization

Processing data from publicly accessible repositories are exempted from GDPR. The EU regulation on the free flow of non-personal data provides the definitions, boundaries and the context to open, public and non-personal data [39]. Given that the processing does not contain any personal data, that is, if the data processing system can identify no individual student, the processing of such data is permissible. Article 9 refers to the categories of personal data. Clause 26 of GDPR states: "*the principles of data protection should apply to any information concerning an identified or identifiable natural person*". The de-identification of data required prior to opening for processing is referred to in Article 29; "*aggregate the data to a level where the individual events are no longer identifiable, the resulting dataset can be qualified as anonymous*".

LA concerns: Current LA practices anonymize data prior to sharing among LA researchers [40]. Anonymization is increasingly becoming a hardly viable technique since it requires removing all personal identifiers in a dataset [8]. Each dataset must, therefore, be considered on a case-by-case basis and evaluate whether there are identifiers that are linked to an individual. As a code of conduct, pseudonymization, that attributes to an identifiable natural person using additional information, is sufficient according to Article 40:2(d). All the means reasonably likely to be used in the processing should be considered. For e.g., singling out—either by the data controller or by another person—that checks if a person can naturally be identifiable. However, pseudonymization is tricky due to the presence of data from multiple sources [41]. Specifically, in the cases of sensitive personal data (Recital 51); special categories of personal data (Recital 52); sensitive data in the health and social sector (Recital 53); sensitive data in the public health sector (Recital 54); data of public interest or interest of authorities for objectives of recognized religious communities (Recital 55); and data on political opinions of data subjects (Recital 56), the processing is prohibited, unless exempted by Article 9:2: Processing of special categories of personal data [41].

Furthermore, according to Recital 26, it must also determine whether there are identifiers that could be used in the future by any third party, including researchers and students etc. to identify individual students because personal data is not just identifiable by single data elements alone, but eventually becomes identifiable once combined or linked with other data. Here, the rights and the conditions for the performers of data processing given under Article 28 is relevant. If a third party is involved in processing, it "*shall be governed by a contract*" according to Article 28:3(b), and such a contract "*ensures that persons authorized to process the personal data have committed themselves to confidentiality or are under an appropriate statutory obligation of confidentiality*".

### 4.6. Transparency in Automated Decision Making

Data-driven methodologies often use automated decision making, especially since LA uses machine learning classification algorithms [3]. Typical classification algorithms receive data in an attribute–value format, where individual data subjects provide values to the attributes. These data are often processed in a black box—that is, the machine learning algorithm does not provide the information on how the decision is being made, rather

the decision itself (the model, or the outcome). However, if the data include personal data, GDPR restricts processing such data when the process of decision making cannot be transparent. Article 12 describes the legal ground of observing transparency in the complete act of processing personal data. Data subjects are empowered, at any point of processing their personal data, to request details on how their data is processed and how conclusions are derived in a "*concise, transparent, intelligible and easily accessible form, using clear and plain language*" according to Article 12:1. Recital 58 defines the principle of transparency as providing "*any information addressed to the public or to the data subject be concise, easily accessible and easy to understand, and that clear and plain language*". If personal data is used for automated decision making, the subject has the right to know the "*existence of automated decision making*" with "*meaningful information about the logic involved, as well as the significance and the envisaged consequences of such processing for the data subject.*" (Article 15:1(h))

### 4.7. Profiling

Categorizing or grouping the data subjects based on certain attributes is known as profiling, and Article 22 refers to automated data processing resulting in the profiling of the data subject. According to 22:1, the data subject shall have the right not to be subjected to a decision based solely on automated processing, including profiling, in likely events that such decisions produce legal effects or similarly significantly affect him or her. If some of the students are labelled as "likely to fail" or "underperformers" or similar, because of an automated process, data subjects have the right to request more information on the decision-making process or can demand to opt-out from such process even at a later stage. Prior to processing data, explicit consent of the data subject for this specific purpose must be obtained (Recital 71). The existence and the nature of profiling must be explicitly stated in the agreement or the consent form, i.e., "*the data subject should be informed of the existence of profiling and the consequences of such profiling*" according to Recital 60. Profiling actions contain highly sensitive information, which would require higher security protocols pursuant to Article 32. Recital 71 further explains the fair and transparent processing; "*use appropriate mathematical or statistical procedures for the profiling*", and, measures to minimize and correct any errors; "*secure personal data in a manner that takes account of the potential risks involved for the interests and rights of the data subject, and prevent, inter alia, discriminatory effects on natural persons on the basis of racial or ethnic origin, political opinion, religion or beliefs, trade union membership, genetic or health status or sexual orientation, or processing that results in measures having such an effect*" (Recital 71).

### 4.8. Data Security

Information security is an explicit legal requirement pursuant to Article 32 and its importance cannot be overemphasized. By Article 32:1(a), it is required to "*implement appropriate technical and organizational measures to ensure a level of security appropriate to the risk*", with "*pseudonymization and encryption of personal data*". Security measures in processing and storing data with "*confidentiality, integrity, availability and resilience of processing systems and services*"; and the possibility to quickly access data in the case of error fixing or meeting the demands of a data subject, as well as regular checking of the security of data is vital (Article 32:1). Thus, educational institutions may even be held to a higher level of data protection standards than private sector organizations. The more sensitive the data, the higher the level of protection. Restrictions enforce the processing of data of special categories by Article 23:1 [42]. Furthermore, according to Article 23:2 [42], specific provisions are required to store, process and share data of these categories.

## 5. Implications of GDPR on LA and the Way Forward

Sections 3 and 4 of this paper separately brought in the current LA research and the GDPR guidelines relevant to data processing, respectively. Therefore, the outcomes of these two chapters combine into some counterarguments such as:

Personalized learning vs. data anonymisation: where GDPR promotes non-disclosure of personal data, and personalization of learning requires learners to be identified individually for relevant interventions.

Automated decision making vs. transparency: where GDPR emphasizes the need for transparency in decision making, while predictive models, in general, is a black box where the process of how outcomes are achieved is not transparent.

Identifying students at risk vs. profiling: identification of dropouts and students at the risk of dropping down is profiling by definition, which is prohibited under GDPR.

Monitoring student behaviour vs. privacy: monitoring the student's behaviour in the learning environment, as well as outside could easily fall into the level of invasion of privacy of the student and would expose student's societal and behavioural stances.

Big data/multimodal data vs. pseudonymization: in which LA promotes aggregating data from many sources for enhanced decision making, while GDPR argues, although data are anonymized to secure the personal information of the subjects, aggregation of data disclose individuals.

### 5.1. Data Anonymisation

An important application of LA is to count on "*new*" information discovered as a result of processing student data to perform "*relevant interventions*" to student learning [1,2,5]. The word cloud of LA literature presented in [43], for example, shows interventions as a prominent keyword. As an essential part of interventions, the individual data subjects (students) need to be explicitly identified, algorithmically or manually, which deteriorates the meaning of anonymization of data [44]. As [8] mentions, de-identification of data is a step forward in the process of anonymization of data and refers to all the identifiable measures of individuals after aggregating data from many sources. This is a way out of GDPR, but it is a challenging and mathematically expensive process. The same article further questions how useful the de-identified data is, meaning that a well de-identified dataset might not be usable for enhancement of personalized learning, and hence, anonymized data does not showcase the full power of LA. Therefore, in occasions of collecting personal data for LA purposes, the explicit consent of the data subject is sought.

### 5.2. Data Minimalisation

LA practices that follow a data-driven approach consist of algorithms and methods to discover "*new*" information from student data aggregated by many sources [45]. In principle, data-driven methods harvest all the data available prior to allowing machine learning algorithms to discover which data provides more accurate information. Therefore, which data is essential for building LA models is unforeseen prior to performing LA. As described above, this contradicts with the GDPR statements of "*data minimalisation*"—collecting data to only serve for the purpose based on informed consent from the data subjects [6]. Thereby, at least the types of LA that use data-driven approaches are subject to a trade-off between the need and the possibility of collecting data from extended sources.

### 5.3. Policies for Data Controllers

GDPR urges explicit descriptions of LA purposes and explicit mention of what personal data is being used for what processing, when and how, as described in the previous section. The general data management regulations in universities may not necessarily be adequate to perform LA research and practices [17]. The legal requirement of LA is to be explicit on the "*particular purpose of data processing*" in the university policies and regulations on data management, and whenever a data collection event occurs an explicit consent is sought. However, due to the power imbalance between the data subject and data controller, data collection and management purposes may not be fully explained to the data subjects [37,38]. The limitations are further illustrated in Recital 43 "*a clear imbalance between the data subject and the controller, in particular where the controller is a public authority, and it is therefore unlikely that consent was freely given in all the circumstances of that specific situation*".

Consequently, this relates to the privacy paradox mentioned in [33], which exactly is the hierarchy between the academic institution and the student. In spite of educational institutions establishing data management policies that include clauses for preserving the privacy of student data, e.g., the Stockholm University policy of "*How personal data is handled at Stockholm University*" [46], the priority is given "*to accomplish our mission as a governmental agency and a university, i.e., to provide research, teaching and societal collaboration*". Moreover, it acknowledges that "*only the personal data required to meet this goal will be processed.*" [46]. However, a better clarification of the specific purpose of LA is provided by Edinburgh University [47] with respective guidelines "*Learning Analytics Principles and Purposes*". This set of guidelines describes the purpose of the use of student data, as "*development of our Learning Analytics activities*". Open University, UK has a policy for LA, which states, the scope and the principles as well as the obligations of all the stakeholders in collecting and managing data for the purpose of LA [48].

Student's right to amend their data or request that their data be erased is, however, recognized already by many institutions [46,49]. The general information of how personal data are being collected and processed is typically included in information pages of the institutions and are directed to responsible officials if any student wishes to receive detailed descriptions.

### 5.4. Automated Decision Making with LA

If machine learning algorithms are used to predict the likelihood of a student being dropped out, it means the decision is automatic [2], which brings in the need of referring to Article 22 of the GDPR about automated decision making (including profiling). The data subject shall have the right to opt-out from events that involve decision making solely based on automated processing of data. Labelling students is highly sensitive, which would require higher security protocols pursuant to Article 32 as mentions above, both at the data management regulations-wise and the explicit consent-wise.

Big data in education is contrasting the GDPR provisions with regards to profiling [45]. Although it falls out of the scope of this article, serious ethical issues may arise as a result of such practices [33]. For instance, assume a college administration has used anonymized data of the past students to learn a model for student performance. This model, for instance, concluded that students of a specific ethnic background who live in a certain geographical zone and score a certain way on standardized tests are highly likely to earn a low grade in an important course. Although personal data is not used for the model building that directly identifies individuals, there is a chance that the college uses the predictive model in making the decisions of recruiting new students and give the least priority to students applying from such categorized regions. Such practice may not violate the legal conditions since decision making on student recruitment in academic institutions typically is performed without external influences, but such knowledge might not be available for university decision makers unless LA is used. However, these circumstances would lead to indirect profiling, which may not typically come to the surface of attention and is hardly avoidable in the current academic practices.

### 5.5. Monitoring Student Behaviour

Monitoring student behaviour in the classroom or online settings, including discussion forums and social media channels, maybe overlooked as an event of an invasion of privacy. Studies based on video-recorded data of how students behave in the classroom and group activities [24] is one such example. If the students who participated in the study have already provided consent for such studies, legal implications cannot be raised. However, monitoring student behaviour require detailed descriptions at the stage of getting the consent as mentioned above, and the access controls in such an event should be documented in the policy. Access to the data containing student behaviour in and outside the classroom may be limited and restricted (e.g., not every lecturer should have access but rather the program director or a designated officer should have access),

as described under pseudonymisation in Article 28. However, deriving new information from students' behaviour data increases the risk of attributing data to a natural person [44], which challenges the anonymity of the data subjects [6,10,37].

It is also important to consider what happens to a student because of the profiling. Will they be forced to take extra classes? Will the students be labelled as incompetent as a consequence of taking extra classes? Will this impact their grade or the potential to obtain a letter of recommendation? Such questions must be answered before implementing legally and ethically correct practices of supporting students by monitoring their behaviour.

### 5.6. Who Is the Beneficiary of LA?

Personal data and the lawful processing of data should solely be based on the legitimacy of the interest and the need for data subjects. However, the typical assumption that the students be the primary beneficiary of LA practices, doesn't hold always [33]. Ref [43] shows that about 65% of LA research in Higher Education focuses on the researchers and administrators, where about 40% of it is just about them alone. This shows that the primary beneficiary of LA is not the data subjects (students), and the teachers are of the lowest focus. In some situations, students might be an indirect beneficiary, but the current trend directly serves the administration and researchers, according to [43]. Thus, the question would be why the student data are used for benefiting other entities than students. However, such discussion is out of the scope of this study.

### 5.7. Special Concerns on Real-Time Analytics

Real-time analytics is becoming increasingly popular since it provides the opportunity for timely interventions for students at risk [1]. When the live LA is performed, a substantial portion of the collection and processing of data will be automated, leaving less chance to screen data prior to the processing. Although it is challenging to decide exactly what real-time data would appear and what the consequences of analytics would be, a clear and transparent process that data subjects are aware of, with clear documentation on the kind of data involved in the processing (e.g., personal and sensitive data), the purpose of the processing, the lawful grounds of the processing (e.g., consent); transparency of the process, and students' rights to access the data and opting out from analysis is mandatory for a GDPR friendly LA.

### 5.8. Provisions for Processing Personal Data

Data controllers may lawfully use the provision that "*processing is necessary for the performance of a task carried out in the public interest or in the exercise of official authority vested in the controller, or on grounds of the legitimate interests of a controller or a third party*" according to Recital 69. However, the data subject still has the right to object processing "*relating to his or her particular situation*". Thus, "*It should be for the controller to demonstrate that its compelling legitimate interest overrides the interests or the fundamental rights and freedoms of the data subject*" (Recital 69). Specifically, education institutions have a primary responsibility to support the students to perform well in their education. Thus, in such circumstances, the GDPR general conditions of processing personal and sensitive data could be overridden by clear and transparent regulations by institutions on the use of data.

### 5.9. Path to Sustainable LA with the Lawful Use of Data Comply with GDPR

In summary, for LA practices to comply with GDPR, several measures are needed to be taken from the stage prior to data collection to interventions for learning as resulted by this study.

#### 5.9.1. The Pathway to GDPR Friendly Data for LA

Based on the abovementioned discussions, and the outcome of the side-by-side comparison of LA practices and GDPR provisions, a promising pathway to lawful LA can be

defined which may allow LA to sustain itself in the presence of GDPR. Figure 5 summarises all these facts together and presents a simple pathway to lawful LA.

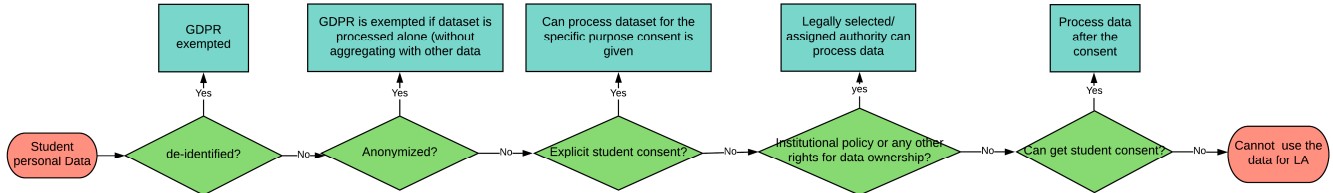

**Figure 5.** Pathway to assurance of data comply with GDPR.

5.9.2. Legal Checkpoints

For the pathway in Figure 5 to be functional, it is important to be reminded about the checkpoints at each of the stages in the pathway, which is summarised below.

De-identified data—if a dataset, irrespective of its harvested from a single source or multiple sources, is fully de-identified for its sensitive elements, such can be used to perform any LA tasks. For de-identification.

- Methods and measures should be stated in an institutional LA policy;
- Assure confidentiality, integrity, availability and resilience of processing and systems (Article 32:1).

Anonymization of personal data (and with respect to sensitive data)—exempted from GDPR limitations if a dataset is processed alone (without aggregating with data from other sources). Thus, with respect to anonymisation:

- The data subject should not be identifiable (Article 29);
- Pseudonymized data should be authorised by the controller according to a legal policy (Article 40);
- Sensitive data need special reservations (Recital 51).

Explicit consent from students—The data can be used to perform LA only for the specific purpose that the consent is given. In this regard, the contract should include:

- The explicit purpose is transparent to the student (Article 12);
- If there is profiling involved, it is explicitly described in the contract (Article 22);
- If automated decision making is involved or not (Article 15);
- For how long the data is retained for processing and how the data controller commits for the confidentiality of data (Article 28).

Institutional policy or any other rights for data ownership—Legally selected/assigned authority can process data. In such situations:

- Institutional (LA) policy should include how institutions use student data. Provide information to the data subject on how data is processed (Article 15);
- Transparent on the rights of the students to request more information on how their data are processed and how the results are used (Article 15 and Recital 63).

Student consent can be obtained retrospectively—In the event of recognising the need for LA activities after the learning activities occur, such data could still process after the consent from the students and/or under the conditions of data ownership. It is important to be reminded about:

- Genuine interest of the data subjects (Recital 46);
- Freedom to be excluded (Recital 2);
- Transparency in the process. Entering into a contract (Recital 40);
- The purpose is explicitly stated in the contract (Recital 33);
- Freedom of data subjects to revert their consent (Article 7:3);

In general, the following conditions should also be important in performing LA activities complied by GDPR:

- Confidentiality, integrity, availability and resilience of processing and systems (Article 32:1);
- Security and privacy checking of personal data by data controllers (Article 23);
- Student right to know if and how personal data is processed (Article 15);
- Student right to know the logic behind results of processing (Recital 61);
- Student right to lodge complaints (covered in many articles and recitals);
- Student right to edit and change personal data (covered in many articles and recitals).

## 6. Concluding Remarks

This study has focused on the dilemmas of standard practices of LA vs. the GDPR rules. Accordingly, it is argued that LA research requires a more concrete understanding of the potential and limitations regarding the use of personal data to conduct LA. New insights to a deeper understanding of the legal implications of GDPR towards LA is essential for legitimate propagation of the field. The outcome of the close investigation of the LA practices against the GDPR resulted in that if student personal data are de-identified after data from different sources are aggregated, processing such data does not have limitations under GDPR. Every other occasion and practice, explicit student consent is required. When the data are automatically collected, the student should be informed duly if such data are intended to be processed. If students are being profiled in the process of data processing, such practice should be transparent and reversible.

However, the technological implication of de-identification of student data is still a question to be addressed. The amount of data accumulating in systems is growing exponentially [10], raising the question of whether the data is "*de-identified enough to waive*" the GDPR requirement. From a technical point of view, the event of revocation of consent, and how such an event affects LA are the other questions to be sought out in sustainable LA.

Sharing data with third parties and other countries, which is the main concern of Article 44–50 and respective recitals, is also something essential to concern if LA to be sustainable and reusable. This should be carefully read and followed if the data controllers are external entities. This study also excluded systematically investigating existing institutional and other relevant policies to identify the gaps, which would also be another relevant and promising further work.

**Funding:** This research was funded by Stockholm University Rector's fund (SU FV-1.1.9-0314-15). The project team, especially the legal expert Liane Colonna is highly acknowledged for the valuable contributions.

**Institutional Review Board Statement:** Not applicable.

**Informed Consent Statement:** Not applicable.

**Data Availability Statement:** Not applicable.

**Conflicts of Interest:** The authors declare no conflict of interest.

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
