# Peer review of "For Learning Analytics to Be Sustainable under GDPR—Consequences and Way Forward"

_sustainability, doi:10.3390/su132011524_

Round 1
Reviewer 1 Report
The paper “For Learning Analytics to be sustainable under GDPR – consequences and way forward” describes the impact of GDPR on the use of learning analytics in higher education. The subject of this paper is very relevant for higher education and the learning analytics community. The author refers to the law articles and recitals and their influence on the use of learning analytics in education and the LA research. They draw out the implications of GDPR on Learning analytics in a clear list. They propose a pathway to assurance of data comply with GDPR in a clear schematic presentation. This also gives this paper an important practical value for the lecturers. The paper has a good structure and is clearly written.
The author has done a literature study by automated mining and using text mining methods. For comparison the analysis of only LAK publications was done. As a result two word clouds images are presented. Text mining seems a very interesting and effective approach to analyze literature where there are a lot of publications. Unfortunately the resulting word clouds are no deep representation and give very little insight in the subject studied. It is not clear to me how this literature study is used further in the paper. Thus it is also not clear what is the benefit of using such method. In particular because the literature review / discussion further seems to be done in a traditional way. I suggest to elaborate this more. The literature study presents relevant literature, including review articles and research papers. Considering the field of Learning analytics which is highly in development it is necessary to include also some more recent literature. I also suggest to include the work by Drachsler and Greller from 2016 which had a huge impact later on in this field “Privacy and Analytics – it’s a DELICATE Issue. A Checklist for Trusted Learning Analytics” that in my opinion is relevant for the work presented in this paper and comment on it.
References 5 and 22 are not complete/correctly cited and need correction.
Author Response
Dear reviewer,
Many thanks for the feedback.
Point 1: The author has done a literature study by automated mining and using text mining methods. For comparison the analysis of only LAK publications was done. As a result two word clouds images are presented. Text mining seems a very interesting and effective approach to analyze literature where there are a lot of publications. Unfortunately the resulting word clouds are no deep representation and give very little insight in the subject studied. It is not clear to me how this literature study is used further in the paper. Thus it is also not clear what is the benefit of using such method. In particular because the literature review / discussion further seems to be done in a traditional way. I suggest to elaborate this more.
Response 1: Many thanks for the valuable comment. The text mining has been conducted for understanding the context of A, but not necessarily to establish deep, specific, and concrete findings. The reason is that the main interest of the study is to see what are the topics addressed in the field, to position the paper in the context of GDPR provisions. However, section 2 is edited clarifying this.
Point 2: The literature study presents relevant literature, including review articles and research papers. Considering the field of Learning analytics which is highly in development it is necessary to include also some more recent literature. I also suggest to include the work by Drachsler and Greller from 2016 which had a huge impact later on in this field “Privacy and Analytics – it’s a DELICATE Issue. A Checklist for Trusted Learning Analytics” that in my opinion is relevant for the work presented in this paper and comment on it.
Response 2: many thanks and appreciate the reference suggestions. I agree that legal issues are very much overlaying in the privacy and ethical aspects, and are very sensitive issues. Therefore this paper only focused on the legal issues, and the other two was considered as out of the scope of the paper.
Point 3: References 5 and 22 are not complete/correctly cited and need correction.
Response 3: many thanks! The references are corrected!
Reviewer 2 Report
I carefully read the paper and found that the topic of the paper is important and the paper is well-written. The authors carefully discussed the issue in details and thoughtfully provided their valuable viewpoints. I do not have any specific comments or suggestions for the authors to improve the paper.
Author Response
Many thanks for the review! Very much appreciated!
Reviewer 3 Report
Thank you for the opportunity to review this interesting article. However, I found some ambiguities regarding:
1. Abstract. The author does not clearly present the purpose of this article and does not present the results obtained from its realization. I suggest the author to do this!
2. Literature review. This section presents a meta-data analysis of LA practices. My opinion is that this should have been mentioned in the section dedicated to research methodology so that readers can understand much more clearly the methodology used by the author!
3. Conclusions. The author does not clearly highlight the conclusions obtained from the study!
As readers cannot clearly understand the structure of the article, the author is asked to do so by indicating whether this study is a meta-data analysis and to specify the research methodology used in a section specifically dedicated to this!
Author Response
Dear reviewer,
Many thanks for the feedback.
Point 1: 1. Abstract. The author does not clearly present the purpose of this article and does not present the results obtained from its realization. I suggest the author to do this!
Response 1: Many thanks for the valuable comment. The abstract is edited and the purpose is explicitly presented.
Point 2: 2. Literature review. This section presents a meta-data analysis of LA practices. My opinion is that this should have been mentioned in the section dedicated to research methodology so that readers can understand much more clearly the methodology used by the author!
Response 2: Thank you for helpful comment. A separate methodology section is added.
Point 3: 3. Conclusions. The author does not clearly highlight the conclusions obtained from the study!
As readers cannot clearly understand the structure of the article, the author is asked to do so by indicating whether this study is a meta-data analysis and to specify the research methodology used in a section specifically dedicated to this!
Response 3: many thanks! The outcome of the study is highlighted in section 5, since the outcomes are descriptive. Refined the section for clarity.
Round 2
Reviewer 3 Report
Thanks to the author for making all my suggestions related to improving the content of the article!
Author Response
Comments and Suggestions for Authors:Thanks to the author for making all my suggestions related to improving the content of the article!
Response to the reviewer:
Many thanks for your excellent suggestions which helped to increase the quality of the manuscript.